# Conceptual Archetype Decomposition for Interpretable and Generalizable Model Decisions

## Abstract

Traditional concept decomposition methods have made significant progress in improving the interpretability of deep learning models, but they still face many challenges. A key issue is that they often lack traceable explanations for concepts, making it difficult to understand and verify how models make decisions and provide explanations based on specific concepts. To overcome this limitation, this paper proposes a new method—Conceptual Archetype Decomposition (CAD)—which aims to provide more interpretable concept learning and decision-making process. Unlike existing methods, our approach ensures that each concept can be represented as a linear combination of training samples, with its total activation value equal to 1. This constraint limits the learning space of the concepts and enhances their interpretability. Therefore, the advantage of our method lies in its fine-grained concept activation decomposition, which directly constructs the explanatory space between training samples and concepts. Through a dual-index decision mechanism, we deeply analyze the relationship between test samples and training samples. Extensive experiments on the CUB and ImageNet datasets demonstrate that our model not only improves decision transparency but also exhibits stronger generalization ability in multi-class classification tasks. Our code is available at: https://anonymous.4open.science/r/CAD-4510/.

## 1 Introduction

In recent years, the increasing reliance on deep learning models in high-risk domains such as healthcare and autonomous driving has highlighted the urgent need for model interpretability (Wang & Chung, 2022; Corfmat et al., 2025). While these models have achieved impressive performance, their opacity poses a significant challenge to understanding the decision-making process (Ribeiro et al., 2016; Selvaraju et al., 2017), which is crucial for ensuring their reliability and safety. Specifically, the lack of clear explanations for the features or concepts driving model predictions often undermines the interpretability of the decision-making process. This issue has sparked a wave of research on explainable deep learning models, particularly those based on concepts (Kim et al., 2018; Lee et al., 2024), aiming to extract human-understandable features from complex models.

A promising approach in this area is Concept Recursive Activation Factorization (CRAFT) (Fel et al., 2023), which decomposes the activations of neural network intermediate layers into a set of concept vectors and their corresponding coefficients via Non-Negative Matrix Factorization (NMF) (Lee & Seung, 1999). Although CRAFT provides a framework for identifying latent concepts in the decision process, it suffers from two key limitations: (i) the extracted concepts lack clear interpretability; (ii) it is difficult to establish an effective link between test samples and training samples in the decision process. Specifically, the concept vectors extracted by CRAFT lack clear semantic meaning, making it challenging to directly map them to interpretable features, and as a result, understanding the specific meaning of each concept becomes difficult. Additionally, the relationship between the concepts in CRAFT and the input data is not well-defined, affecting the model's robustness and generalization capabilities.

To overcome these limitations, we propose a novel framework—Concept Archetype Decomposition (CAD)—designed to enhance the interpretability and robustness of concept-based deep learning models. Unlike previous methods such as CRAFT, CAD modifies the standard matrix factorization approach by representing the concept extraction process as a linear combination of the activation matrix $A$ and two additional matrices: the concept index matrix $C$ and the concept reconstruction matrix $B$. This method ensures that each concept is associated with specific training samples during the training process and is formed

through their linear combination, thus avoiding the issue in CRAFT where there is no direct connection between the activation matrix and concept vectors, leading to clearer semantic meanings for each concept.

CAD satisfies a nested bilayer convex combination, meaning that each reconstruction can be fully indexed without encountering the issue in CRAFT where some samples cannot be explained. Additionally, CAD inherently adopts a Low-Entropy Structure, which avoids the influence of hyperparameters when setting regularization loss functions (Zhu et al., 2024). This allows for direct sparse results, thereby mitigating the impact of hyperparameter randomness on the concept decomposition.

During the testing phase, CAD uses the same weight matrix $C$ learned during training to optimize and obtain the concept reconstruction matrix for the test samples, thus enabling the decomposition of new samples into concepts. Extensive experiments on benchmark datasets such as CUB (Wah et al., 2011) and ImageNet demonstrate (Deng et al., 2009) that our method not only enhances decision transparency but also exhibits stronger generalization capabilities. The main contributions of this paper are summarized as follows:

- We propose Concept Archetype Decomposition (CAD), a novel concept extraction method that avoids the distributional mismatch issue present in CRAFT decomposition (Fel et al., 2023), while also ensuring full interpretability, meaning that the concept vectors themselves are interpretable and indexable.

- The CAD design satisfies a nested bilayer convex combination and inherently adopts a Low-Entropy Structure, which helps avoid the influence of hyperparameter randomness in concept decomposition.

- We demonstrate the effectiveness of CAD on benchmark datasets, showing that it outperforms existing methods in both interpretability and robustness. We also provide a fully open-source code package for community research.

## 2 Related Work

### 2.1 Traditional interpretability methods

There are two main directions in the field of interpretability research: one is Post-hoc interpretability, represented by works such as Grad-CAM (Selvaraju et al., 2017), IG (Sundararajan et al., 2017), Shapley (Lundberg & Lee, 2017), and LIME (Ribeiro et al., 2016). The other is the construction of inherently interpretable models, such as Concept Bottleneck Models (CBMs) (Koh et al., 2020). In most cases, Post-hoc methods are more valuable because there has already been a significant body of work that performs exceptionally well, and we only need to provide reasonable explanations for their behavior. On the other hand, inherently interpretable models are subject to the limitation of being "inherently interpretable," which may result in slightly inferior performance compared to models without such constraints.

In this paper, we focus on Post-hoc methods. Existing research, such as Grad-CAM (Selvaraju et al., 2017), IG, and other more advanced attribution methods (Zhu et al., 2024), aims to construct a heatmap to help humans understand which regions are important and what features the model relies on for decision-making. However, the issue with these methods is that they only highlight important regions but fail to explain what is within those regions, specifically what factors contribute to the prediction outcome.

### 2.2 Research and Application of Concept Decomposition Explainability

In recent years, several concept-based decomposition methods have been introduced to address the issue of explaining decision-making criteria. A substantial body of research has demonstrated the applications of concept decomposition in interpretability. For instance, methods like Representational Similarity Via Interpretable Visual Concepts (RSVC) (Kondapaneni et al., 2025) and Representational Difference Explanations (RDX) (Kondapaneni et al., 2025) use concept decomposition results to study differences between models. Modeldiff (Shah et al., 2023) leverages this technique to investigate the impact of different training strategies, making the study of concept decomposition interpretability highly valuable. A representative method, Testing With Concept Activation Vectors(TCAV) (Kim et al., 2018), allows users to provide datasets with and without a specific concept, from which it learns a concept vector to decompose the corresponding concept. This approach also enables observation of factors present in an unseen dataset through concept matching. However, this method relies on manually provided concept sets, making it unsuitable in cases where the data is complex or the number of concepts is large. Additionally, since concept selection depends on human expertise, it faces the challenge of missing important concepts due to incomplete concept sets.

CRAFT (Fel et al., 2023), another concept extraction method, utilizes non-negative matrix factorization to decompose intermediate feature vectors (Lee & Seung, 1999). However, its training distribution and decomposition distribution are inconsistent, leading to the introduction of many out-of-distribution concepts during decomposition. Furthermore, CRAFT requires feature vectors to be strictly positive, making it unsuitable for models that do not meet this condition.

The core issue with the above decomposition methods lies in the fact that the concept vectors are learned, and obtaining interpretability for these concept vectors is inherently difficult. In other words, while we can decompose concepts, we cannot be certain of what those concepts represent. The proposed CAD method leverages convex properties to ensure that each concept can be indexed to its corresponding samples, thereby providing full interpretability. We ensure that every sample's decomposition can necessarily lead to its concept composition, and each concept is directly associated with the corresponding sample explanation.

## 3 METHOD

In this paper, we propose a novel framework, Conceptual Archetype Decomposition (CAD), aimed at enhancing the interpretability and robustness of concepts in concept-based interpretable deep learning models. Compared to previous methods such as CRAFT (Fel et al., 2023), our approach addresses two key issues: (i) the lack of intrinsic explanations for the extracted individual concepts, and (ii) the challenge of establishing associations between test and training samples during the model's decision-making process. The following sections will present our method in detail, organized into three subsections: Problem Definition, Preliminary, and Conceptual Archetype Decomposition.

### 3.1 PROBLEM DEFINITION

Given a training sample $x^{(i)} \in \mathbb{R}^{w \times h \times 3}$ (where 3 corresponds to the image's RGB channels, and $h$ and $w$ are the image's height and width, respectively) and its corresponding class label $y^{(i)}$, we aim to design a method that decomposes the feature representations of deep neural networks into human-understandable concepts. Traditional methods for concept decomposition typically rely on subregion cropping (**?**), where the original training sample $x^{(i)}$ is cropped into a patch $x'^{(i)} \in \mathbb{R}^{c \times s \times s}$ of length and width $s \times s$. However, this approach often results in a shift in the input distribution between model training and testing, increasing the risk of reduced interpretability and robustness of the extracted concepts. This is because the model was not trained on these samples during training, which means that their distribution differs significantly from that of the training set. We train the model by feeding the entire image into the input, ensuring consistent inputs between training and inference. Next, we use a mapping function $g$ that maps the full-image input $x^{(i)} \in \mathbb{R}^{w \times h \times 3}$ to an intermediate state $A(x^{(i)}) \in \mathbb{R}^{w \times h \times c}$ for decomposition (Goodfellow et al., 2016). Finally, the classifier $h$ maps the intermediate state to the output, ensuring $f(x^{(i)}) = h(g(x^{(i)}))$. At this point, $f : x^{(i)} \rightarrow y^{(i)}$.

### 3.2 PRELIMINARIES

To enhance the interpretability of deep neural network models, the CRAFT method (Fel et al., 2023) employs Non-negative Matrix Factorization (NMF) techniques (Lee & Seung, 1999). NMF is an algorithm that decomposes a non-negative data matrix into two non-negative matrices, with each matrix representing different feature dimensions of the data. In CRAFT, NMF is used to factorize the activation matrix from the intermediate layers of the neural network into two matrices, which represent the concept dictionary and the concept coefficients, respectively. In this way, CRAFT is able to identify and extract the latent concepts involved in the decision-making process of the neural network, thereby aiding in the understanding of how the network reasons from input to output. The objective of NMF, in this context, is to minimize the following optimization problem:

$$\hat{U}, \hat{W} = \min_{U,W} \|A - UW^\top\|_F^2, \quad \text{if and only if } U, W \geq 0 \tag{1}$$

First, we need to reshape the matrix so that $A \in \mathbb{R}^{nwh \times c}$ represents the activation matrix extracted from the intermediate layer of the neural network, used to represent the sample. $n$ is the number of samples, and $wh \times c$ is the feature dimension of each sample. $W \in \mathbb{R}^{k \times c}$ represents the concept dictionary matrix, which provides the concept activation vector. $U \in \mathbb{R}^{nwh \times k}$ is the concept coefficient matrix, which contains

the representation of each sample based on these concepts. $k$ represents the number of extracted concepts. Furthermore, CRAFT's (Fel et al., 2023) reconstruction process is $A \approx UW$. Therefore, $U$ and $W$ do not establish a direct connection with the activation $A$, making it difficult to extract the inherent meaning of the concepts. In other words, even if we obtain the final optimized concept matrix $\hat{W}$, we cannot accurately deconstruct it and cannot fully understand the meaning of each concept.

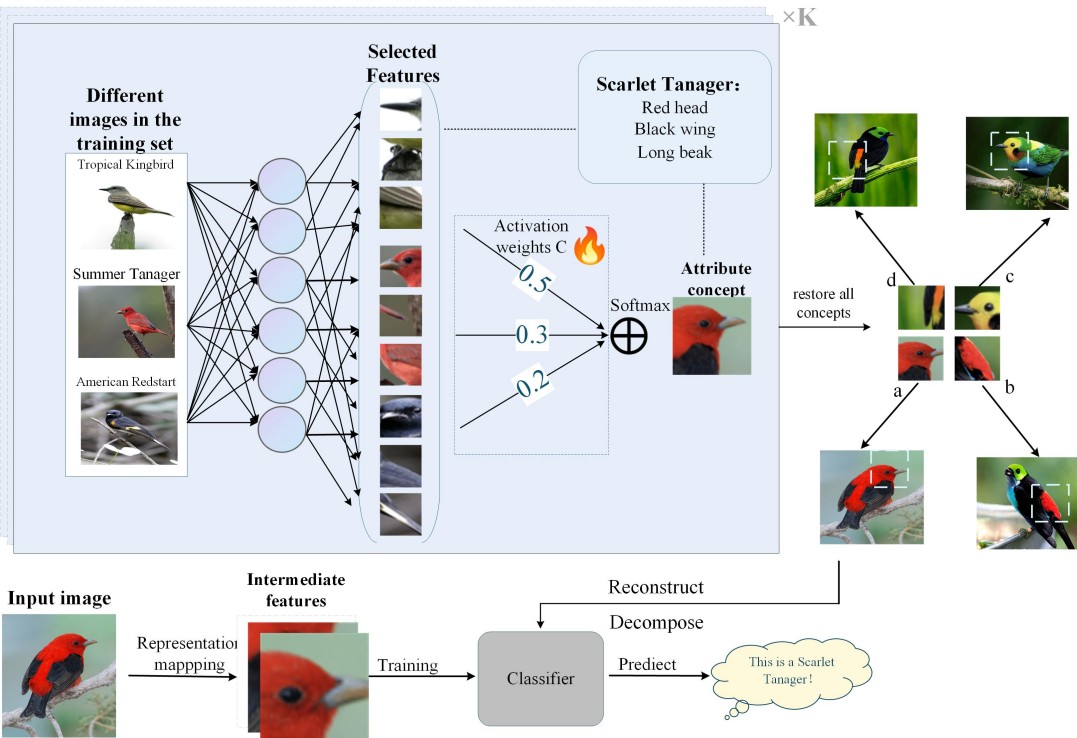

Figure 1: The overall architecture of our proposed Conceptual Archetype Decomposition (CAD) method. The process includes feature extraction from an input image, followed by decomposition and reconstruction, ultimately leading to a classification prediction. The upper panel illustrates how concepts are formed from features selected from a diverse training set and then used to analyze other images.

## 3.3 CONCEPTUAL ARCHETYPE DECOMPOSITION

### 3.3.1 CONCEPT EXTRACTION AND INTERPRETATION

Although the CRAFT method proposes extracting concepts through NMF (Lee & Seung, 1999) to enhance model transparency, its biggest challenge lies in concept interpretability. Because the concept dictionary matrix $W$ generated during NMF decomposition is an optimization artifact, it lacks inherent meaning or intuitive interpretation. To understand the meaning of a specific concept, it is often necessary to sample activation values and extract corresponding images for visualization. However, activation values alone do not directly reflect the specific content of the concept. For example, there may be images with high activation values that do not exclusively represent a specific concept. For example, suppose we sample three images with activation values of 1.6, 1.1, and 0.5, representing varying degrees of activation for the same concept. However, this simple sorting of activation values is insufficient to provide a clear conceptual interpretation, as high activation values do not always imply that the image's content fully aligns with the extracted concept. For example, an image with an activation value of 1.6 may contain a mixture of multiple concepts, rather than simply the salient features of that concept. Therefore, this approach fails to clearly define the specific boundaries and meaning of a concept, and the activation threshold for sampling lacks standardization and controllability. This makes concept interpretability ambiguous and unreliable.

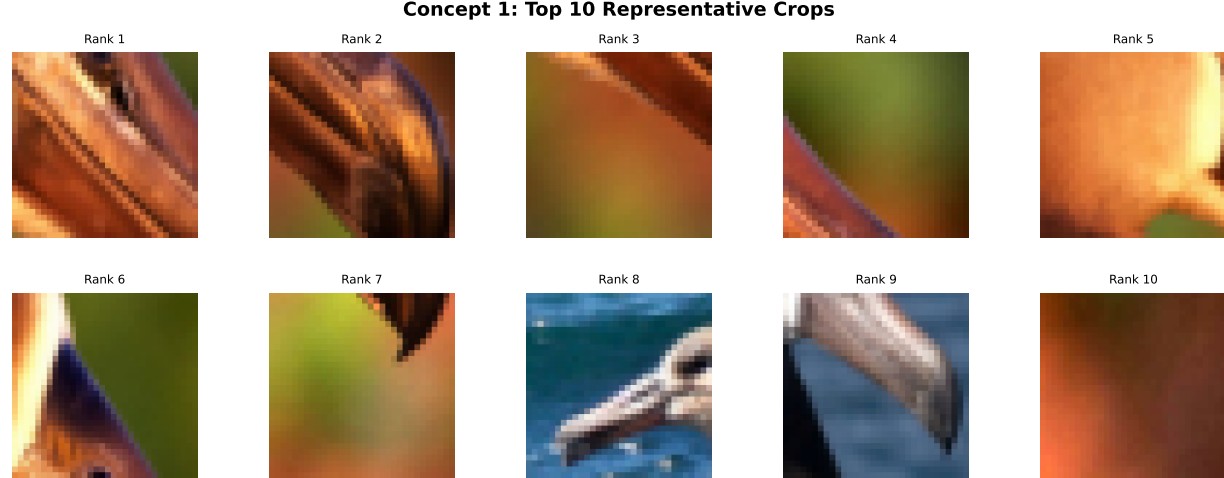

Figure 2: Visualization of conceptual archetype decomposition on CUB dataset: example 1

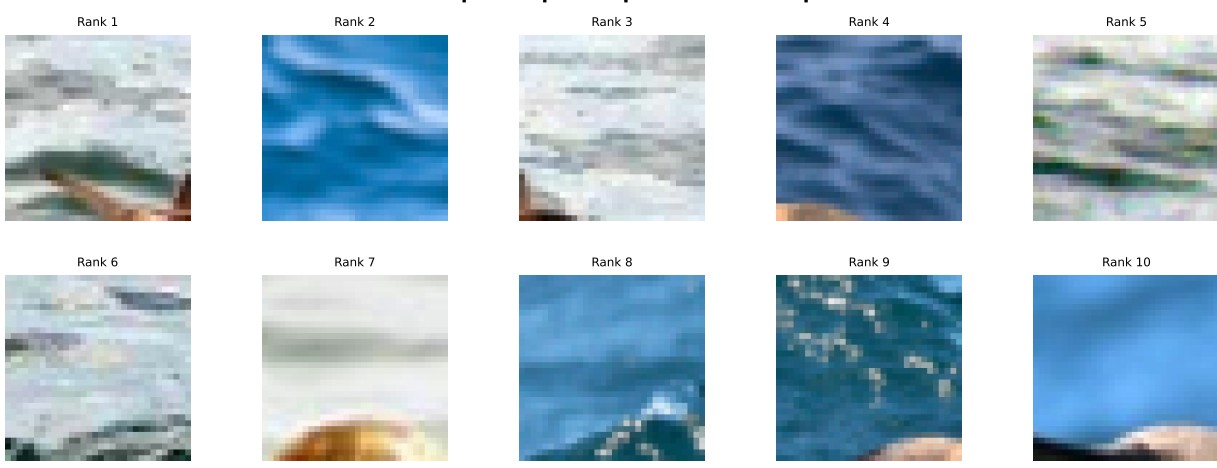

Figure 3: Visualization of conceptual archetype decomposition on CUB dataset: example 2

In this section, to provide a traceable intrinsic explanation for the concepts extracted during matrix decomposition and to establish connections between test and training samples, as shown in Figure 1, we represent the concept reconstruction process as $A \approx A^\top C^\top B$ and call this method CAD, where $A$ is the activation matrix containing the intermediate layer features of the sample and used to represent the sample, $C \in \mathbb{R}^{k \times nwh}$ is the concept index matrix, and $B \in \mathbb{R}^{nwh \times k}$ is the concept reconstruction matrix. Specifically, assuming that the activation matrix we extract from the intermediate layer of the neural network is $A \in \mathbb{R}^{nwh \times c}$, we hope to decompose the concept represented by $z$ into a linear combination of the activation matrix $A$ and the concept index matrix $C$, in the form $z = A^\top C^\top$. To ensure that the concepts extracted during training can be traced back to the training samples, we need to add a restriction $\forall i, \sum_{j=1}^{nwh} C_{ij} = 1$ to the concept index matrix $C$. To ensure that this restriction is strictly true, we perform softmax normalization on the second dimension of the $C$ matrix in each calculation so that the sum of each column in the matrix is 1:

$$C_{ij} = \frac{e^{c_{ij}}}{\sum_{j=1}^{nwh} e^{c_{ij}}} \tag{2}$$

This means that for the elements $C_{ij}$ in the concept index matrix $C$, they play a one-to-one corresponding role with the feature activation $A$ of the intermediate layer of the sample in the construction of the concept $z$. Therefore, our concepts are completely obtained by linear combinations of samples, and any concept can find its constituent feature activations and their weights. At the same time, because the feature activations are spatially associated with the original features (Selvaraju et al., 2017), we can find the image location information of the explanation and use it as the explanation. This also means that since $z$ is strictly composed of the activation matrix $A$ and the concept index matrix C, when we need to get the explanation of the concept, we can directly find the corresponding activation matrix composition through $C$ and find the samples behind it as the explanation. CRAFT needs to use samples for sampling to obtain samples with higher activation values as the explanation of the concept, but this also means that this explanation of the concept is not absolutely accurate. In Figure 2 and 3, we visualize the index sources of Concept 1, which are extracted from different training set image crops, ordered by their index values $C_{ij}$ from high to low.

### 3.3.2 BUILDING ASSOCIATIONS BETWEEN CONCEPTS AND ORIGINAL SAMPLES

In addition to the concept index matrix $C$, we also introduce a concept reconstruction matrix $B$, which is used to reconstruct concepts, with $\forall i, \sum_{j=1}^{k} B_{ij} = 1$. $B$ reconstructs samples by controlling the activation level of each concept. Specifically, $B$ indicates which concepts comprise the sample. We use the same operations as for the concept index matrix $C$. Figure 4 shows the training process of our CAD method.

Here we want to compare the matrix $B$ with the activation matrix $U$ in CRAFT. The $B$ matrix has two obvious advantages: 1. Unified dimensionality. In the demonstration $U$, the activation values of different concepts cannot be compared. For example, the value of a certain feature dimension $U$ is between 0-5, while another may be between 0-1. The different dimensions also mean that we cannot determine the activation of a concept by the value of U. 2. $U$ represents the activation value of the concept, while $B$ represents "composition". Because the sum of the second dimension in the matrix $B$ is constrained to be fixed at 1, we can ensure that the reconstructed activation value is completely composed of the concepts in $Z$, while activation values can only be compared with concepts. In summary, the goal of our CAD optimization is:

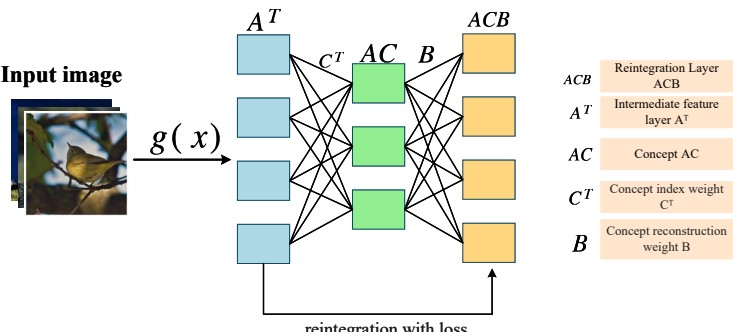

Figure 4: The architecture of our proposed Conceptual Archetype Decomposition (CAD) method.

$$\hat{B}, \hat{C} = \min_{\hat{B}, \hat{C}} \|A - A^{\top} C^{\top} B\| \quad \text{s.t.} \ \forall i, \sum_{j=1}^{nwh} C_{ij} = 1, \ \forall i, \sum_{j=1}^{k} B_{ij} = 1 \tag{3}$$

Next, we need to analyze the properties of CAD. CAD satisfies nested double-convex combination, which is easy to prove:

$$A_i \approx \sum_{j=1}^{k} B_{ij} Z_j = \sum_{j=1}^{k} B_{ij} \left( \sum_{\ell=1}^{nwh} C_{j\ell} A_\ell \right) = \sum_{\ell=1}^{nwh} \left( \sum_{j=1}^{k} C_{j\ell} B_{ij} \right) A_\ell. \tag{4}$$

That is,

$$A^{\top} C^{\top} B \subseteq \text{conv}(Z) \subseteq \text{conv}(A) \tag{5}$$

Where cone(X) represents the convex hull of the original data. For ease of understanding, we give the definition of the convex hull here:

$$\text{conv}(X) := \left\{ \sum_{i=1}^{n} \alpha_i x_i \ \middle| \ \alpha_i \geq 0, \ \sum_{i=1}^{n} \alpha_i = 1 \right\} \tag{6}$$

Here, $X = \{x_1, x_2, \ldots, x_n\} \subset \mathbb{R}^d$. This property also means that any reconstruction can be fully indexed. However, PCA and CRAFT often project points outside the data convex hull, resulting in unrealistic and difficult-to-interpret combinations. This can easily lead to interpretation crises in scenarios requiring strict interpretability. Our CAD, however, provides a rigorous theoretical guarantee against this.

Another important property of CAD is its inherent low-entropy structure. This means that it naturally produces sparse matrices without the need for a regularized loss function. This avoids the influence of hyperparameters when setting regularized loss functions and prevents randomness from affecting the confidence of the interpretation of the results. This can be explained from the perspective of the degrees of freedom of the matrices. Due to the constraint that the sum is unity, the degrees of freedom of the $B$ matrix are $\Delta^{k-1}$, and the degrees of freedom of the $a$ matrix are $\Delta^{nwh-1}$. This is then simply proved using Carathéodory's theorem (Carathéodory, 1907). In practice, data distributions often approximate the ground-dimensional manifold, so the sparsity can be even greater than the upper bound proven by Carathéodory's theorem.

CAD also has a very advantageous property: **archetypes converge to the extreme points of the convex hull of the data.** Intuitively, a pole cannot be written as a convex combination of two other points in the convex hull. That is, in convex analysis, a point $x \in \mathcal{C} = \text{conv}(X)$ is a pole if and only if:

$$x \neq \lambda x' + (1-\lambda)x'', \quad \forall x', x'' \in \mathcal{C} \setminus \{x\}, \ \lambda \in (0,1) \tag{7}$$

This also means that the decomposition of concepts will select representative samples, and when reconstructing samples, representative concepts will be selected. Concepts and reconstructed samples have natural atomicity, which can be easily proved using the Cutler & Breiman theorem Cutler & Breiman (1994) combined with the convex hull property.

Finally, in the interaction phase, if we want to obtain a unified conceptual explanation, we only need to fix $Z$, which is $A^\top C^\top$ during training, and use Eq. 3 to optimize the new B matrix to obtain a unified explanation.

## 4 EXPERIMENT

### 4.1 EXPERIMENTAL SETUP

In this paper, we design and implement multiple experiments to validate the effectiveness and superiority of the proposed Conceptual Archetype Decomposition (CAD) method. We use two widely adopted benchmark datasets in visual tasks: CUB (Wah et al., 2011) and ImageNet (Deng et al., 2009), and select two common deep learning models as backbones: NF_ResNet50 (Brock et al., 2021) and VIT-B/32(Dosovitskiy et al., 2021). Specifically, the CUB dataset contains images of 200 bird species, providing rich fine-grained label information, making it suitable for testing the model's performance in fine-grained object classification. The ImageNet dataset, with 1,000 object categories, has a large-scale data set, making it ideal for evaluating the model's robustness and generalization ability in large-scale visual classification tasks. To comprehensively assess our CAD method, we choose two competitive baseline methods: CRAFT (Fel et al., 2023) and PCA (Jolliffe, 1986). The CRAFT method, which extracts concepts based on Non-negative Matrix Factorization, serves as a direct comparison to our method, while PCA, as a classic dimensionality reduction method, is used as another baseline to demonstrate the differences in model interpretability and performance between various concept extraction methods.

### 4.2 EXPERIMENT 1: VALIDATION OF CONCEPT RECONSTRUCTION ERROR

To evaluate the reconstruction error of concepts on both the training and test sets, we propose a validation method based on Mean Squared Error (MSE). A smaller MSE indicates that the reconstructed features closely match the original features. The results of this experiment are presented in Table 1. This

experiment probes whether *CAD*'s convex-combination design—$\hat{A} = A^\top C^\top B$ with per-column simplex constraints on $C$ and $B$—translates into *in-distribution* reconstructions at test time. The key signal is the generalization gap $\Delta = \text{MSE}_{\text{test}} - \text{MSE}_{\text{train}}$ (the "Variation" column), not the raw training MSE whose magnitude is confounded by backbone feature scales. Across all datasets/backbones, CAD yields gaps that are *approximately* $5.9\times$–$12.9\times$ smaller than CRAFT and $3.3\times$–$12.8\times$ smaller than PCA (e.g., CUB/NF-ResNet50: 24.73 vs. 156.35/170.78; CUB/ViT-B/32: 0.26 vs. 1.54/0.87; ImageNet/NF-ResNet50: 0.08 vs. 1.03/1.02; ImageNet/ViT-B/32: 0.12 vs. 1.35/0.79). This pattern matches the method's inductive bias: since $\hat{A} \in \text{conv}(A)$, CAD reconstructs test activations as convex mixtures of training activations and avoids *off-hull* extrapolation. In contrast, dictionary–coefficient factorizations (e.g., NMF-style $UW$ in CRAFT) can aggressively minimize train error yet reconstruct test features outside the training hull, inflating $\Delta$ despite tiny training MSE (e.g., CUB/NF-ResNet50: 1.87 $\rightarrow$ 158.21). Put differently, CAD trades a bit of training fit for *distribution-matched* test reconstructions—a bias–variance choice enforced by the simplex/low-entropy structure of $B$ and $C$, which acts as an implicit regularizer. The smallest gaps appear on the larger ImageNet and the stronger ViT-B/32 backbone, suggesting that the convex-hull constraint scales favorably as the feature geometry becomes richer (consistent with archetypes concentrating near extreme points of $\text{conv}(A)$). Finally, these results empirically corroborate the critique of cropping-based decompositions: when training/decomposition inputs are mismatched (sub-regions vs. full images), test reconstructions drift; CAD's full-image mapping maintains train/test alignment.

Table 1: Validation table of Concept Reconstruction Error (Train vs. Test). We report mean $\pm$ std and *Variation* = Test$-$Train. The smallest *Variation* per dataset is bolded.

| | Model | CUB / NF-ResNet50 | | | CUB / ViT-B/32 | | |
|---|---|---|---|---|---|---|---|
| Variant | Dataset | Train | Test | Variation | Train | Test | Variation |
| Ours | CUB | $86.74 \pm 22.37$ | $111.48 \pm 24.03$ | **24.73** | $1.14 \pm 0.17$ | $1.40 \pm 0.16$ | **0.26** |
| CRAFT | CUB | $1.87 \pm 0.46$ | $158.21 \pm 36.66$ | 156.35 | $1.01 \pm 0.06$ | $2.55 \pm 0.09$ | 1.54 |
| PCA | CUB | $5.56 \pm 1.10$ | $176.33 \pm 40.48$ | 170.78 | $0.93 \pm 0.30$ | $1.80 \pm 0.21$ | 0.87 |

| | Model | ImageNet / NF-ResNet50 | | | ImageNet / ViT-B/32 | | |
|---|---|---|---|---|---|---|---|
| Variant | Dataset | Train | Test | Variation | Train | Test | Variation |
| Ours | ImageNet | $0.89 \pm 0.18$ | $0.97 \pm 0.19$ | **0.08** | $1.46 \pm 0.16$ | $1.58 \pm 0.16$ | **0.12** |
| CRAFT | ImageNet | $0.10 \pm 0.02$ | $1.14 \pm 0.19$ | 1.03 | $1.14 \pm 0.07$ | $2.49 \pm 0.10$ | 1.35 |
| PCA | ImageNet | $0.19 \pm 0.03$ | $1.20 \pm 0.20$ | 1.02 | $1.06 \pm 0.29$ | $1.85 \pm 0.20$ | 0.79 |

### 4.3 Experiment 2: Validation of Concept Reconstruction Classification Accuracy

This evaluation asks a stronger question than Exp. 1: not only must the reconstruction be close to the original features, it must also be *decision preserving*. We therefore classify *only from reconstructed features* and sweep the number of concepts $k$ (Table 2, Fig. 5). Two signals matter: (i) the *level* of accuracy after reconstruction and (ii) its *stability* as $k$ varies.

*Level.* CAD attains near-ceiling accuracy across datasets/backbones (typically 99%–100%), while CRAFT/PCA trail substantially—by +30–65 points on CUB and +5–13 points on ImageNet. This aligns with CAD's convex-hull bias: the mapping $\hat{A} \in \text{conv}(A)$ keeps test features inside the training support, so a Lipschitz classifier (e.g., a linear head) experiences at most a bounded logit perturbation proportional to $\|A - \hat{A}\|$; hence decision regions and margins are largely preserved. In contrast, CRAFT's free dictionary and PCA's global subspace can push reconstructions *off-hull*, which degrades separability even when training error is small (See §4.2).

*Stability.* CAD's mean $\pm$ std bands are narrow and essentially *flat* in $k$; CRAFT/PCA exhibit large dispersion (std up to $\approx 40$ on CUB) and non-monotone slopes, indicating a "$k$-lottery" effect where adding concepts changes the factorization in ways that disrupt the classifier. By constraining $B, C$ to simplices, CAD enforces sparse, low-entropy mixtures that vary smoothly with $k$, acting as an implicit regularizer that keeps decision geometry consistent.

*Backbone/dataset trends.* Gaps shrink on ImageNet and ViT-B/32 because the base representations are already highly linearly separable; nevertheless CAD retains a consistent edge and the tightest variability, suggesting that the convex-combination mechanism scales favorably as the feature geometry becomes richer. On CUB, where features are more brittle and class margins are thinner, CAD's no-extrapolation property is most beneficial, yielding the largest accuracy gains and the smallest variance.

Table 2: Validation of concept reconstruction classification accuracy (%). We report mean ± std across seeds; higher is better. Best per column is in **bold**.

| #Concepts $k$ | CUB / NF-ResNet50 | | | CUB / ViT-B/32 | | | ImageNet / NF-ResNet50 | | | ImageNet / ViT-B/32 | | |
|---|---|---|---|---|---|---|---|---|---|---|---|---|
| | Ours | CRAFT | PCA | Ours | CRAFT | PCA | Ours | CRAFT | PCA | Ours | CRAFT | PCA |
| 10 | **99.5** ± **5.4** | 71.8 ± 38.0 | 34.5 ± 40.3 | **99.6** ± **2.5** | 43.1 ± 40.1 | 65.6 ± 34.5 | **99.9** ± **0.8** | 94.4 ± 19.4 | 95.0 ± 19.0 | **97.6** ± **8.8** | 88.6 ± 20.5 | 90.7 ± 19.5 |
| 20 | **99.5** ± **4.3** | 74.9 ± 35.4 | 44.6 ± 38.8 | **99.8** ± **1.8** | 44.7 ± 38.5 | 64.7 ± 31.8 | **99.9** ± **0.9** | 95.1 ± 17.4 | 93.5 ± 20.0 | **98.5** ± **5.1** | 86.3 ± 21.8 | 89.8 ± 16.8 |
| 30 | **99.4** ± **4.9** | 75.8 ± 33.2 | 52.3 ± 36.4 | **100.0** ± **0.3** | 47.7 ± 36.7 | 65.2 ± 29.2 | **99.8** ± **1.0** | 95.2 ± 17.1 | 95.1 ± 15.8 | **99.0** ± **3.9** | 86.5 ± 20.0 | 90.5 ± 12.8 |
| 40 | **99.4** ± **5.4** | 76.2 ± 33.0 | 54.1 ± 36.3 | **99.9** ± **0.5** | 49.1 ± 36.6 | 65.1 ± 29.1 | **99.8** ± **1.3** | 95.0 ± 17.2 | 95.3 ± 12.7 | **99.3** ± **1.9** | 86.0 ± 19.6 | 88.1 ± 16.2 |
| 50 | **99.3** ± **5.3** | 76.6 ± 32.8 | 56.1 ± 34.6 | **100.0** ± **0.0** | 50.9 ± 36.0 | 66.5 ± 27.8 | **99.6** ± **1.2** | 95.1 ± 17.1 | 95.4 ± 13.8 | **99.3** ± **2.9** | 86.2 ± 18.9 | 88.1 ± 16.7 |

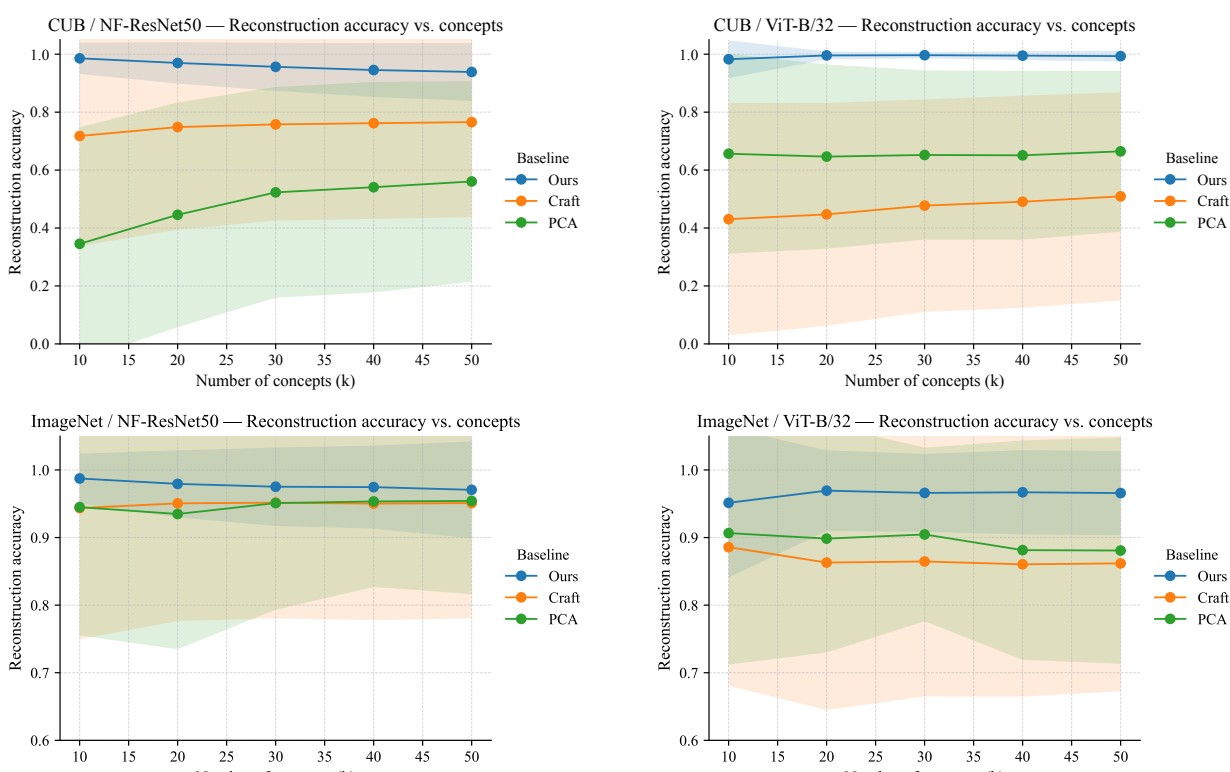

Figure 5: Validation chart of concept reconstruction classification accuracy.

## 5 CONCLUSION

This paper presents a novel concept extraction method—Conceptual Archetype Decomposition (CAD)—designed to enhance the interpretability and robustness of concept-based deep learning models. By introducing a concept index matrix and a concept reconstruction matrix, CAD ensures a direct association between the extracted concepts and the training samples, overcoming the limitations of existing methods (such as CRAFT and PCA) in terms of concept interpretability and generalization ability. Our experimental results demonstrate the effectiveness and superiority of CAD on several benchmark datasets, including CUB and ImageNet. Overall, CAD not only provides more intuitive and reliable concept representations for model interpretability, but also shows stronger performance and robustness in multi-class classification tasks. Our method offers new insights for research on explainable deep learning models, with significant theoretical value and application potential, particularly in high-risk fields such as healthcare and autonomous driving, where enhancing the transparency and safety of model decision-making processes is of critical importance.

ETHICS STATEMENT

We have read and will adhere to the ICLR Code of Ethics. This work uses only public data, involves no human subjects or personally identifiable information, and therefore does not require IRB review. Results are reported for research purposes only; we release anonymized code/configurations to support verification, and will disclose any funding sources and potential conflicts of interest upon acceptance.

REPRODUCIBILITY STATEMENT

To support reproducibility, we release an anonymized repository with all experiment details including training/evaluation scripts, default hyperparameters, configuration files, and software/hardware environment.

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

## LLM Usage Disclosure

We used large language models (OpenAI GPT-4o and GPT-5) as auxiliary tools for grammar checking and language polishing of the manuscript. These models were not involved in research ideation, experimental design, implementation, or analysis. The authors take full responsibility for all content.

