# OpenReview forum: "Conceptual Archetype Decomposition for Interpretable and Generalizable Model Decisions"
_ICLR.cc/2026/Conference — ICLR 2026 Conference Withdrawn Submission_

### Official Review · Reviewer_HYzh · 2025-10-27

**Soundness:** 1
**Presentation:** 1
**Contribution:** 2
**Rating:** 2
**Confidence:** 3

**Summary:**

This paper introduces Conceptual Archetype Decomposition (CAD), a novel inherent framework designed to improve the interpretability and generalization of concept-based models. Existing decomposition approaches, such as CRAFT, utilize Non-Negative Matrix Factorization (NMF) to extract latent concepts from neural activations; however, they often fail to provide traceable or clear semantically meaningful explanations, as their concepts are not explicitly linked to real training samples. CAD learns the concept from samples and ensures that the decomposition concept can be used to reconstruct the original features, as evidenced by the experiment, which shows a lower disparity between the reconstructed errors of training and testing samples, and a lesser sacrifice of classification capability compared to CRAFT.

**Strengths:**

1. The CAD provides an inherent concept decomposition method that aims to improve and offer an intuitive interpretation.
2. The CAD provides a traceable intrinsic explanation for the concept decomposition and bridges the relation between test and training samples.

**Weaknesses:**

1. Although the paper claims that CAD achieves better interpretability, it does not provide experimental evidence to support this assertion.

2. The formulation of $A \approx A^T C^T B$  is confusing, as the dimensions between the left-hand side and the right-hand side appear to be inconsistent. Based on the description, the dimension of left-hand side formulation is $A \in \mathbb{R}^{nwh\times c}$. However, the right-hand side formulation doesn’t match. Besides, how $A^TC^T \in \mathbb{R}^{c \times k}$ dot product with $B \in \mathbb{B}^{nwh \times k}$?

3. The presented concept in Figures 2 and 3 are hardly able to convey the clear semantics, which is also hard to convince that the CAD can provide clearer concept semantics than the CRAFT.

4. It would be helpful if the authors could provide qualitative results demonstrating that CAD supports more interpretable and more clearly semantical concepts compared to CRAFT.

5. The experiment in Table 1 shows the lower concept reconstruction error between the training and testing sets, but the purpose of the experiment isn’t clear.

**Questions:**

1. The formulation in Equation 3 ($A-A^T C^T B$) does not seem to match the symbol in Figure 4 ($ACB$).
2. How is the performance in Experiment 2 calculated? The reported results reach nearly 100% across different models and datasets, which raises curiosity about the evaluation procedure.

---

### Official Review · Reviewer_3Xub · 2025-10-30

**Soundness:** 1
**Presentation:** 1
**Contribution:** 2
**Rating:** 2
**Confidence:** 4

**Summary:**

The paper proposes Conceptual Archetype Decomposition (CAD), a concept-based factorization method defined as
$\hat{A} = A^\top C^\top B$.
Here, $A$ is the activation matrix, $C$ (concept index) ensures each concept $z = A^\top C^\top$ is a convex combination of training activations via per-column softmax normalization, and $B$ (concept reconstruction) expresses samples as convex combinations of concepts.
This nested convex-hull structure allows test samples to be decomposed into training-defined concepts by fixing $C$ and optimizing $B$.

**Strengths:**

+ The paper clearly identifies two limitations of prior work, CRAFT:
(i) extracted concepts lack semantic interpretability, and
(ii) test samples are not explicitly connected to training samples.
CAD’s convex-hull formulation directly targets these issues.
+ The geometric idea of keeping reconstructions inside the convex hull of training activations is elegant and provides a theoretically consistent interpretation space.

**Weaknesses:**

Overall, the paper suffers from serious presentation and formatting issues:
figures are blurry, equations are poorly typeset, page size and headers do not follow the official ICLR template, and the writing quality is well below conference standards.

1. Limited datasets.
Only CUB and ImageNet are used, which are not the typical benchmarks for concept-based interpretability (e.g., Broden, CelebA, AwA).
More fine-grained or domain-diverse datasets are needed.

2. Inadequate baselines.
The paper only compares with PCA and CRAFT.
PCA is a dimensionality-reduction method, not a concept decomposition approach.
Proper baselines should include TCAV, ACE.
It is unclear why CRAFT’s original comparisons (e.g., with ACE) were not reproduced.

3. Mischaracterization of CBMs.
The authors claim that inherently interpretable models necessarily trade off accuracy, which is not universally true, many CBM variants achieve both interpretability and strong performance.

4. Missing comparison with attribution methods.
No qualitative or quantitative comparison with Grad-CAM, Integrated Gradients, or other post-hoc visualization baselines is provided.

5. Poor visualization quality.
The presented decompositions are hard to interpret visually; it is unclear whether the extracted components have meaningful semantic content.

**Questions:**

1. Concept definition on ImageNet:
How are the ImageNet “concepts’’ obtained, and how is the number of concepts (10–50) selected?

2. Use of the same weight matrix C (lines 62–63):
Technically, this forces every test representation to be reconstructed as a convex combination of training samples.
While this prevents OOD extrapolation, it completely removes the ability to represent novel concepts outside the training distribution.

3. Unclear statement at lines 134–135 (“subregion cropping (?)”):
What method is referenced here? Please provide a citation or clarification.

4. Dataset choice:
Why are only two datasets evaluated?
More diverse or fine-grained datasets (e.g., Flowers, Aircraft, or medical concept benchmarks) would better validate the claimed interpretability and robustness.

---

### Official Review · Reviewer_ob9W · 2025-11-01

**Soundness:** 1
**Presentation:** 2
**Contribution:** 1
**Rating:** 2
**Confidence:** 4

**Summary:**

This paper proposes Conceptual Archetype Decomposition (CAD), a method that improves model interpretability by representing concepts as fixed-sum linear combinations of training samples. This approach enhances decision transparency and shows stronger generalization of reconstructed classification performance on two image classification tasks.

**Strengths:**

The intention to make concepts traceable back to its original training sample is good.

**Weaknesses:**

1. Unsupported Claim of Interpretability vs. Evidence: The paper's central thesis rests on achieving a "more interpretable" process. However, this claim is asserted rather than empirically demonstrated. The authors equate the existence of a "traceable" method (concepts as linear combinations of samples) with a "more interpretable" outcome for the user. To be convincing, the paper must provide direct evidence of this interpretability. This could include, for example, qualitative case studies that walk through a test sample's decision, trace it back to the specific training "archetypes," and demonstrate how this "dual-index" explanation provides a more lucid or actionable insight than existing methods. Without this, the primary claim remains unsubstantiated. Fig 3 and 4 do not suffice.

2. Lacking Formal Interpretability Benchmarks: The field of XAI has moved toward formally evaluating interpretability claims. The paper would be significantly strengthened by benchmarking its method against established interpretability metrics. While the method is new, its utility as an explanation is not measured. For instance, the authors could adopt benchmarks similar to those in CRAFT or related works, evaluating concept utility, completeness, or faithfulness. How effective are these concepts at explaining model failures? Can users leverage these "archetypes" to debug the model? By omitting such analyses, the paper misses a crucial opportunity to quantify how and how much more interpretable its method truly is.

3. Lacking contextualization (Automatic Concept Methods): The paper's experimental validation and related work sections are notably missing a key class of competitors: automatic concept extraction methods. The authors should discuss and quantitatively compare CAD against prominent baselines (e.g. ACE). This is a critical omission. It is unclear whether CAD (which seems to decompose into pre-defined or sample-based archetypes) offers a practical advantage over methods that discover concepts automatically from the model's latent space. This comparison is essential for contextualizing the paper's contribution and understanding the trade-offs of the proposed "conceptual archetype" approach. Sparse autoencoders are also highly relevant in reconstructing representations in a more interpretable manner.

4. Lacking Ablation Study: The proposed CAD method involves several key design choices that are presented without justification or analysis. A robust ablation study is necessary to understand the method's sensitivity and to validate these choices. For example:
* Layer Sensitivity: From which layer are the "conceptual archetypes" extracted? How does this choice (e.g., early, middle, vs. final convolutional layers) impact the resulting concepts' quality and the model's downstream performance?
* Architecture Dependence: The experiments are conducted on CUB and ImageNet, but the robustness of CAD across different backbone architectures (e.g., ResNets, Vision Transformers) is not explored. Does the method's effectiveness depend on a specific architectural family? Without these ablations, it is difficult to assess the method's generalizability or understand which components of the framework are most critical to its success.

**Questions:**

See weaknesses. The most significant question is how CAD is more interpretable besides hand-wavey arguments?

---

### Official Review · Reviewer_aYsc · 2025-11-02

**Soundness:** 2
**Presentation:** 1
**Contribution:** 2
**Rating:** 2
**Confidence:** 3

**Summary:**

The authors proposed Conceptual Archetype Decomposition (CAD), a post-hoc framework for improving model interpretability through concept-based analysis. CAD represents each concept as a convex combination of training activations, constrained such that the combination weights sum to one, forming what the authors call a nested bilayer convex structure.
This formulation introduces two matrices: a concept index matrix $C$ that links each concept to its contributing samples, and a concept reconstruction matrix $B$ that reconstructs each sample from the learned concepts.
The authors claim that this convex-hull design naturally induces low-entropy sparsity and improves both interpretability and generalization. Experiments on the CUB and ImageNet datasets reportedly show that CAD achieves lower reconstruction error and higher classification accuracy on reconstructed activations than baseline methods such as CRAFT and PCA.

**Strengths:**

- **S1. Geometrically Consistent and Intuitively Structured Design**

The convex-hull formulation in CAD provides a straightforward way to keep concept representations grounded in the training data.
By requiring that each concept be expressed as a convex combination of existing activations, the method maintains internal consistency and avoids producing out-of-distribution features.
While this idea is conceptually simple, it offers a reasonable geometric intuition for linking concepts to data without requiring architectural changes or complex supervision. The formulation thus gives CAD a clear structure that is easy to interpret, even if its novelty and practical advantages remain modest.

**Weaknesses:**

- **W1. Overstated Theoretical Guarantees about Sparsity and Archetypal Convergence**

CAD’s theoretical section (Sec. 3.3.2) claims that its convex-combination design naturally produces sparse matrices without a regularized loss function, citing Caratheodory’s theorem as justification.
However, this interpretation misuses the theorem’s scope. As established in classical convex analysis [1, 2], the theorem guarantees only that any point in $\mathbb{R}^d$ can be expressed as a convex combination of at most $d + 1$ points from a set.
It is purely existential and does not imply that an optimization procedure will converge to such sparse representations or that learned coefficients will exhibit low entropy.
Since CAD’s $B$ and $C$ matrices are parameterized by softmax activations, their weights are strictly positive and dense unless explicitly regularized.
Without an entropy or $l_1$ penalty, the model cannot ensure the sparsity or low-entropy structure claimed as an inherent property.

The authors also claim that “archetypes converge to extreme points of the convex hull,” referencing Cutler & Breiman (1994).
Yet both this work and later analyses (e.g., [3]) emphasize that archetypes are constrained to lie within the convex hull and typically tend toward its boundary, not the vertices themselves.
Convergence to extreme points occurs only under restrictive geometric conditions (e.g., noiseless or perfectly separable data).
Because CAD’s interpretability depends on equating each concept with an atomic exemplar anchored at a hull vertex, the absence of a theoretical or algorithmic guarantee undermines this key claim.
To strengthen the argument, the authors could explicitly acknowledge the existential--not constructive--nature of Carathéodory’s theorem, and introduce mechanisms (e.g., entropy regularization or simplex-sharpening) that encourage sparse, vertex-proximal archetypes in practice.

- **W2. Weak Semantic Grounding and Foreground–Background Confusion in Concept Visualization**

While CAD claims to produce “conceptual archetypes” that capture human-interpretable object features, the visualizations in Figs. 2 and 3 reveal that many top-ranked crops correspond to background regions or peripheral textures rather than the foreground object (e.g., bird body parts).
This pattern indicates that CAD’s convex-combination mechanism does not distinguish between semantically meaningful regions and incidental correlations in feature space. Because the model operates purely on activation similarity without spatial or saliency constraints, it tends to select high-activation but non-diagnostic features--branches, skies, or color patches--that co-occur with classes but are not intrinsic concepts. Consequently, the resulting archetypes appear statistically coherent yet semantically shallow, weakening the claim that they offer transparent, object-centric explanations.

This limitation contrasts with prior post-hoc concept-based XAI methods, such as [4], [5], and [6], which explicitly assess concept existence, localization, and faithfulness to model decisions.
These methods incorporate tests to verify that discovered concepts align with human-perceptible regions and influence predictions, and they often include cleanup or hierarchy mechanisms to mitigate background bias.
The proposed method, in contrast, reports no localization or intervention analysis and lacks mechanisms to ensure that archetypes reflect genuine object semantics rather than background co-variation. Integrating object-centric priors--such as attention weighting, saliency-guided masking, or class-conditional filtering—would help align CAD’s archetypal decomposition with the interpretability standards established in the concept-based XAI literature.

- **W3. Limited Experimental Design and Incomplete Evaluation Metrics**

The experimental validation of CAD is too narrow to substantiate its claims of interpretability and generalization.
The authors evaluated only on CUB and ImageNet, using two backbones (NF-ResNet50, ViT-B/32) and just two baselines--[5] and PCA. While CRAFT is a fair comparison, PCA is not a concept-based interpretability model and offers little insight into semantic or causal fidelity.
More recent post-hoc concept-based frameworks, including [4], [6], and [7], provide complementary criteria such as concept localization, completeness, faithfulness, and human alignment. Omitting these baselines and metrics makes CAD’s reported gains difficult to interpret within the current explainability landscape.

Moreover, CAD’s evaluation relies solely on reconstruction MSE and classification accuracy of reconstructed features, which assess geometric fidelity but not whether the learned concepts actually drive predictions or align with human-interpretable semantics.
To make its experimental design more rigorous, the authors should adopt faithfulness and intervention metrics such as concept insertion/deletion or TCAV sensitivity tests to validate causal relevance, and report richer diagnostics like concept sparsity, activation entropy, and overlap statistics to empirically verify the claimed “low-entropy” structure.


- **Reference**
- [1] Rockafellar RT. Convex analysis:(pms-28).; Ch.17-(3)
- [2] Boyd S, Vandenberghe L. Convex optimization. Cambridge university press; 2004 Mar 8.; Sec. 2.3.4 explicitly notes that the theorem “provides an upper bound on the number of atoms in a convex combination” and does not guarantee that algorithms yield sparse or vertex-anchored representations.
- [3] Thurau C, Kersting K, Bauckhage C. Yes we can: simplex volume maximization for descriptive web-scale matrix factorization. InProceedings of the 19th ACM international conference on information and knowledge management 2010 Oct 26 (pp. 1785-1788).; Reinforces that archetypes’ positions depend on data geometry and optimization bias; convergence to vertices is not guaranteed without additional constraints.
- [4] Ghorbani A, Wexler J, Zou JY, Kim B. Towards automatic concept-based explanations. Advances in neural information processing systems. 2019;32.
- [5] Fel T, Picard A, Bethune L, Boissin T, Vigouroux D, Colin J, Cadène R, Serre T. Craft: Concept recursive activation factorization for explainability. InProceedings of the IEEE/CVF Conference on Computer Vision and Pattern Recognition 2023 (pp. 2711-2721).
- [6] Vielhaben J, Bluecher S, Strodthoff N. Multi-dimensional concept discovery (MCD): A unifying framework with completeness guarantees. Trans. Mach. Learn. Res.. 2023 Jan 1.
- [7] Kondapaneni N, Mac Aodha O, Perona P. Representational similarity via interpretable visual concepts. InThe Thirteenth International Conference on Learning Representations 2025 Mar 30.

**Questions:**

Most of my main concerns or questions have been outlined in the Weaknesses section.

---

### Note · Authors · 2025-11-14

I have read and agree with the venue's withdrawal policy on behalf of myself and my co-authors.